# OpenReview forum: "SelfGrader: Detecting Jailbreak Attacks on Large Language Models with Token-Level Logit Distribution"
_ICLR.cc/2026/Conference — Submitted to ICLR 2026_

### Official Review · Reviewer_FHc7 · 2025-10-18

**Soundness:** 3
**Presentation:** 2
**Contribution:** 2
**Rating:** 4
**Confidence:** 4

**Summary:**

The paper proposes SelfGrader, a guardrail that converts jailbreak detection into a numerical grading task using NT-based (numeric token) logits and a Dual-Perspective Logit (DPL) score. The method aims to be lightweight (low latency / memory) and robust across diverse attack families without training or keyword matching. Main claims are laid out in the intro and method (ICL anchoring + DPL scoring) and evaluated on Llama-3-8B-Instruct and Vicuna-13B-v1.5 with numerous attacks and ablations.

**Strengths:**

1. Simple, implementation-friendly idea (two forward passes, L=1 default, small logit slice), with clear latency/memory accounting
2. Method clarity: the DPL rule is well specified (normalization, tail-trim w, balance parameter λ), and the ICL prompts are fully disclosed in the appendix.
3. Broad evaluation: many attack families (static, model-specific, adaptive), two targets, and several competitive baselines.

**Weaknesses:**

1. Lack of novelty:
The idea of using NT-based logits as safety signals is merely a direct variation of existing approaches such as Self-Evaluation (Ren et al., 2023) and Self-Certainty (Kang et al., 2025). This paper only applies that idea to jailbreak detection, without offering fundamental innovation. Moreover, the method of extracting values through logits is similar to QGuard: Question-based Zero-shot Guard for Multi-modal LLM Safety.

2. Performance instability:
Looking at Table 1, even SelfGrader (Q=101) shows wildly varying PGRs across attack types — 52.38% on MultiJail but 0% on ActorAttack. While the authors emphasize lower average values, individual attacks still exhibit high bypass rates.

3. Vulnerable to adaptive attacks:
In Table 2, SelfGrader (Q=101) performs worse on the TAP attack (ASR 6.00%, PGR 20.00%) than GradientCuff (ASR 2.00%, PGR 3.67%).

4. Increasing Q is meaningless:
Table 1 shows little difference across Q = 2, 10, 101, 1000 (mean ASR = 1.75%, 1.75%, 1.32%, 1.08%). If performance is nearly identical, the design rationale for increasing the number of numeric tokens (NT) is weak. The claim that “Q=1000 is more precise” is not supported by the numbers.

5. Prompt injection vulnerability:
In the appendix (Table 7), PGR spikes up to 73.66%. This implies that even simple injection attacks can effectively neutralize the guardrail.

The claim that SelfGrader is “robust” because ASR remains 0% under prompt injection is not convincing. Elevated PGR values (up to 73.66%) clearly indicate that the scoring mechanism was successfully manipulated. While no harmful outputs were observed in this limited setting, the fact that the guardrail accepts adversarial prompts as benign is itself a security failure. Moreover, the adversarial prompt tested was simplistic; more sophisticated injection strategies could plausibly yield nonzero ASR. Thus, the evidence suggests SelfGrader is vulnerable to scoring corruption, and robustness claims are overstated without broader evaluations and more realistic adversarial tests.

Table 5.
SelfGrader+SelfDefend improves performance under MultiJail but degrades performance under ActorAttack.
Under what conditions is the combination advantageous, and when does it hinder performance?

**Questions:**

1. Why is there almost no performance difference when increasing Q from 2 to 1000? What theoretical significance does expanding the NT set have?

2. Why are the results unstable in the Vicuna experiments? Is this due to limitations of the NT set, or differences in the model itself? Clarification is needed.

3. Given that the method is essentially defenseless against prompt injection, what methods do you propose to mitigate this weakness?

---

> ### Author Response · Authors · 2025-11-24
> **Response to Reviewer FHc7 (1/4)**
>
> **We sincerely thank Reviewer FHc7 for the thoughtful and constructive review comments. We have responded to each weakness and question in detail.**
>
> > [Weakness 1] Lack of novelty: The idea of using NT-based logits as safety signals is merely a direct variation of existing approaches such as Self-Evaluation (Ren et al., 2023) and Self-Certainty (Kang et al., 2025). This paper only applies that idea to jailbreak detection, without offering fundamental innovation. Moreover, the method of extracting values through logits is similar to QGuard: Question-based Zero-shot Guard for Multi-modal LLM Safety.
>
> **Response:** Thank you for raising this point. Although these works (Self-Evaluation (Ren et al., 2023), Self-Certainty (Kang et al., 2025), and QGuard (2025)) also reference logits, the way they use them, and the representation space in which they operate, are entirely different from SelfGrader. We will have the following discussed in our revised version.
>
>    1. **Difference from Self-Evaluation and Self-Certainty:** These methods (not designed for safety evaluation) rely on full-vocabulary logits, in which not all of these logits are appropriate for safety judgment. A naïve strategy would be to inspect the logits of safety-related keywords only. However, relying on keyword logits is fundamentally unreliable: the keyword space is open-ended, highly context-dependent, and impossible to define comprehensively for robust maliciousness or benignness assessment. In contrast, numerical tokens (NTs) form a closed, invariant, and task-aligned label space. Under the SelfGrader prompt, the model internally maps its hidden state to a scalar score, making the logits over NTs a direct and high-SNR (signal-to-noise ratio) readout of the model’s safety judgment (either maliciousness or benignness).
>
>    2. **Difference from QGuard:** QGuard uses the logits of the semantic tokens “yes” and “no” to build a binary classifier, which inherently cannot provide continuous scoring or support dual-perspective reasoning. That is to say, it does not operate in numerical space:
>       - QGuard does not use numerical tokens;
>       - It does not map the model’s internal safety judgment to a scalar score;
>       - It does not permit symmetric maliciousness–benignness analysis.
>
>         In contrast, SelfGrader introduces a fundamentally different scoring mechanism, i.e., a logit-based numerical readout that extracts a continuous safety score, enabling richer and more fine-grained safety evaluation than binary classification.

---

> ### Author Response · Authors · 2025-11-24
> **Response to Reviewer FHc7 (2/4)**
>
> > [Weaknesses 2&3]
> W2: Performance instability: Looking at Table 1, even SelfGrader (Q=101) shows wildly varying PGRs across attack types — 52.38% on MultiJail but 0% on ActorAttack. While the authors emphasize lower average values, individual attacks still exhibit high bypass rates.
> W3: Vulnerable to adaptive attacks: In Table 2, SelfGrader (Q=101) performs worse on the TAP attack (ASR 6.00%, PGR 20.00%) than GradientCuff (ASR 2.00%, PGR 3.67%).
>
> **Response:** Thank you for the insightful comments. We address these concerns below.
>
> 1. **Regarding performance variations across attack families.**
>    1. Different jailbreak families manipulate the model’s internal activations in fundamentally different ways, leading to distinct attack patterns. The MultiJail attack leads to a substantial rise in PGR for many methods. For instance, Llama Guard, WildGuard, and GuardReasoner (Post) each exhibit PGR values exceeding 70%, and SelfDefend with safety-alignment training should have a better decision boundary for MultiJail attack, but still performs poorly with PGR reaching 72.38%. This degradation stems from the fact that these approaches reply on generating textual judgments or safety statements. Such outputs are vulnerable to linguistic ambiguity, stylistic variation, decoding randomness, and prompt sensitivity, which are further amplified under jailbreaks using low-resource language like MultiJail.
>    2. In contrast, our method maintains PGR under 52.38% under this challenging setting. More importantly, it can be seamlessly paired with safety-aligned models. For example, while SelfDefend alone is not robust to these attacks due to the aforementioned biases, incorporating our method remains effective. This is because numerical tokens (NTs) form a closed, invariant, and task-aligned label space, which is less affected by low-resouce languages. In result, the SelfGrader(+SelfDefend) achieve 0% in ASR and 4.76% in PGR on MultiJail attack.
>
> 2. **Clarification on the TAP results.** For TAP attack, GradientCuff reports ASR = 5.00% and PGR = 10.00%. The values ASR = 2.00% and PGR = 3.67% correspond to the average over three adaptive variants. Under the same averaging results, SelfGrader (Q = 101) achieves ASR = 2.67% and PGR = 8.33%. While GradientCuff shows +0.67% better in ASR, it requires white-box gradient access and is more than 21× slower. Depending on the deployment scenario and resource constraints, either method may be preferable.
>
> 3. **No silver bullet.** No guardrail achieves uniformly low PGR across all attack types while keeping efficient, because different attacks exploit different weaknesses. Safety methods must balance robustness, speed, and access assumptions, creating an inherent tradeoff. This is the key motivation, as well as contribution, of SelfGrader: offering fast, gradient-free, and model-agnostic safety assessment with SOTA overall defense performance.
>
> ---
>
> > [Weakness 4 & Question 1] W4: Increasing Q is meaningless: Table 1 shows little difference across Q = 2, 10, 101, 1000 (mean ASR = 1.75%, 1.75%, 1.32%, 1.08%). If performance is nearly identical, the design rationale for increasing the number of numeric tokens (NT) is weak. The claim that “Q=1000 is more precise” is not supported by the numbers. Q1: Why is there almost no performance difference when increasing Q from 2 to 1000? What theoretical significance does expanding the NT set have?
>
> **Response:** Thank you for the thoughtful comments. We first clarify that the size of the numeric token set in size of $Q$ is not the core design choice of SelfGrader. The central idea of our method is the logit space evaluation pipeline and the use of an invariant numeric label space. $Q$ only controls the granularity of these numerical tokens (NTs) space rather than the safety mechanism itself. With this context, the observations in Table 1 become expected and we explain below.
>
> 1. The role of $Q$ is not to change model predictions, but to discretize logits at different resolutions. Thus, small variation in the averaged numbers is an expected property, not a design weakness.
> 2. Expanding $Q$ increases the resolution of the NT space and reduces discretization error when mapping model’s internal safety judgment to discrete NTs. This improves the precision and smoothness of the safety scores, especially in threshold sweeping and calibration. These improvements are not always visible in a single averaged ASR number but become important when tuning the operating point in real deployments.
> 3. Since $Q$ is not the core mechanism, relatively small values (such as $Q$ = 101) already work well. Larger $Q$ provides marginal but consistent gains in score stability and calibration, while introducing only **negligible latency overhead**. Smaller $Q$ values remain competitive due to the simplicity of the NT space.

---

> ### Author Response · Authors · 2025-11-24
> **Response to Reviewer FHc7 (3/4)**
>
> > [Weaknesses 5&6 & Question 3] W5: Prompt injection vulnerability: In the appendix (Table 7), PGR spikes up to 73.66%. This implies that even simple injection attacks can effectively neutralize the guardrail. W6: The claim that SelfGrader is “robust” because ASR remains 0% under prompt injection is not convincing. Elevated PGR values (up to 73.66%) clearly indicate that the scoring mechanism was successfully manipulated. While no harmful outputs were observed in this limited setting, the fact that the guardrail accepts adversarial prompts as benign is itself a security failure. Moreover, the adversarial prompt tested was simplistic; more sophisticated injection strategies could plausibly yield nonzero ASR. Thus, the evidence suggests SelfGrader is vulnerable to scoring corruption, and robustness claims are overstated without broader evaluations and more realistic adversarial tests. Q3: Given that the method is essentially defenseless against prompt injection, what methods do you propose to mitigate this weakness?
>
>
> **Response:** Thank you for raising this point. We address the concern and report additional evaluations below,
>
> 1. **Distinct threat models.** Prompt injection and jailbreak correspond to fundamentally different threat models. Our defense mechanism is designed specifically for the jailbreak setting defined in our system. Therefore, Table 7 should not be interpreted as an evaluation of SelfGrader against general-purpose prompt injection attacks. The goal of Table 7 is to illustrate that under naïve injection attempts, the adversary cannot simultaneously bypass both the target LLM and the guardrail.
> 2. **Not trivially vulnerable to scoring corruption.** SelfGrader is not easily influenced by superficial scoring corruption. Its stability is reflected in safety neutral settings such as math tasks in GSM8K and code tasks in HumanEval shown in Figure 2. In these evaluations, numurical text perturbations do not cause incorrect safety decisions. Both tasks yield zero percent false positive rate. These observations show that the scoring mechanism is not trivially disrupted by prompt level scoring corruption.
> 3. **Practical mitigation strategies.** If stronger protection against prompt injection is required in deployment, practitioners can incorporate additional scoring criteria in the prompts in Appendix B.1 and B.2.
> 4. **Additional evaluations on prompt injection attack.** Even without any dedicated prompt injection design, our method shows resilience against stronger prompt injection. We randomly sample 300 adversarial queries from a stronger prompt injection benchmark BIPIA [1] and evaluate all guardrails. under the same setting. The results are summarized below:
>
> [1] Benchmarking and Defending Against Indirect Prompt Injection Attacks on Large Language Models. KDD 2025.
>
> | Guardrails / ASR, PGR  | BIPIA (ASR ($\downarrow$) / PGR ($\downarrow$) in \%) | Latency (Sec.) | Extra Memory (MB)
> |--------------------------------------------------|-------|-|-|
> | LLama-3-8B-Insturct (No Defense) | 1.33/- |- |-|
> | Perplexity Filter | 1.33/100.00 |0.09 | 13112.46|
> | GradSafe | OOM | -|-|
> | GradientCuff | 0.33/39.33 | 10.49|827.33|
> | Prompt Guard | 0.00/0.33  |9.22|1077.71
> | Llama Guard (Pre) | 1.33/100.00 |0.22 | 13571.94|
> | Llama Guard (Post) | 1.33/100.00 |0.22 | 13598.72|
> | SelfDefend (Direct) | 0.00/19.33 |0.40 | 13113.04|
> | SelfDefend (Intent) | 0.33/67.66 |1.26 | 13116.22|
> | WildGuard (Pre) | 0.66/81.66 |1.58 | 27861.87|
> | WildGuard (Post) | 0.66/89.00 |1.49 |27882.30 |
> | GuardReasoner (Pre) | 1.00/89.33 |8.58 |15414.55 |
> | GuardReasoner (Post) | 1.00/97.66 | 8.22| 15414.60|
> | **SelfGrader (Q=101)** | 0.00/5.33 | 0.30| 328.91|
>
>
> Although SelfGrader is not designed for prompt injection, it shows strong resistance to the prompt injection attacks in BIPIA benchmark. Compared with other LLM-based detectors such as GradientCuff, Llama Guard, SelfDefend, WildGuard, and GuardReasoner, SelfGrader reduces PGR from the 39.33–100% range down to only 5.33%, improving robustness by roughly 85–95%. Classification-based method Prompt Guard remains a strong baseline with a low 0.33 percent PGR, but it incurs noticeably higher computational cost. In contrast, SelfGrader delivers similar accuracy benefits while being more resource-efficient, reducing extra memory consumption by over 95 percent (328 MB versus 1.07 GB) and lowering latency from 9.22 sec for Prompt Guard to 0.30 sec, a 30$\times$ speed-up.

---

> ### Author Response · Authors · 2025-11-24
> **Response to Reviewer FHc7 (4/4)**
>
> > [Weaknesses 7 & Question 2] W7: Table 5. SelfGrader+SelfDefend improves performance under MultiJail but degrades performance under ActorAttack. Under what conditions is the combination advantageous, and when does it hinder performance?
> Q2: Why are the results unstable in the Vicuna experiments? Is this due to limitations of the NT set, or differences in the model itself? Clarification is needed.
>
> **Response:** Thank you for the thoughtful questions.
> 1. In our setup, SelfGrader relies on the model that provides the logits used for mapping NTs. Its effectiveness therefore depends on how harmful and benign behaviors are separated in the logit space of that model. When we switch from the target LLM to the SelfDefend model as the logit provider, the safety-aligned training of SelfDefend reshapes this separation pattern.
>
>     The combination becomes advantageous when SelfDefend’s alignment improves how harmful and benign behaviors are separated in its logit space, making this boundary clearer for a given attack family. This occurs in MultiJail. Although SelfDefend alone is not fully robust to the low-resource language artifacts in MultiJail, its safety supervision still pushes harmful intent into a more distinguishable region than the target LLM does. SelfGrader can leverage this clearer separation to achieve stronger performance. Conversely, when the alignment-induced separation is less compatible with an attack’s characteristics, the combination may hinder performance.
>
> 2. Regarding the instability in the Vicuna results, the variability arises from both the NT set and the characteristics of Vicuna’s own representation space. First, Vicuna exposes only a small subset of unique numeric tokens (0–9) without additional numerical variants. This limits the scalability of NT mapping when no auxiliary model is used, because SelfGrader must compress the logit space into a much coarser set of numeric anchors. With fewer NTs, perturbations from certain attacks can lead to larger relative fluctuations in the mapped scores.
>
>     Similarly, the stability of SelfGrader also depends on how harmful and benign behaviors are separated in the model’s logit space. Vicuna’s fine-tuning procedure and training data produce a separation pattern that differs from the Llama-3-8B-Instruct model used in Table 1. For certain attack families, the harmful–benign contrast in Vicuna’s logits is weaker or less consistently organized, which makes the NT mapping more sensitive to small perturbations. Therefore, the observed instability is not caused by inherent limitation of the SelfGrader design, but arises from the interaction between NT resolution and the way Vicuna internally structures harmful and benign signals.
>
>     Fortunately, because SelfGrader is model-agnostic and can be paired with any logit provider, both challenges can be addressed by exploring improved alignment auxiliary model whose logit separation better matches the threat model of interest.

---

### Official Review · Reviewer_b1Lx · 2025-10-27

**Soundness:** 3
**Presentation:** 3
**Contribution:** 2
**Rating:** 2
**Confidence:** 4

**Summary:**

This paper presents a lightweight guardrail called SelfGrader for detecting jailbreak attacks against large language models (LLMs). The proposed method, SelfGrader, reframes jailbreak detection as a numerical grading task using token-level logits rather than generated text or auxiliary classifiers. Specifically, the model is prompted to “grade” the maliciousness of a query using a compact set of numerical tokens and uses the logit distribution as an evaluation metric.
SelfGrader introduces two key innovations 1, An in-context learning (ICL) prompting strategy that curated query–score examples over time. And 2, A Dual-Perspective Logit (DPL) scoring rule that combines both maliciousness and benignness assessments to yield a stable and balanced safety score.
Experiments across multiple jailbreak benchmarks and benign datasets show that SelfGrader achieves competitive or superior detection accuracy compared with state-of-the-art guardrails while maintaining low false positive rates.

**Strengths:**

1, Strong empirical performance

Experiments across diverse jailbreak and benign benchmarks show consistently low attack success rates and false positive rates compared to stronger baselines.

2, Dual-perspective scoring for robustness

The proposed Dual-Perspective Logit (DPL) is interesting, it seems to effectively balance maliciousness and benignness evaluations, reducing both false positives and false negatives.

3, Lightweight and efficient design

SelfGrader operates without external classifiers, gradient computations, or fine-tuning, achieving low latency and minimal GPU overhead, making it practical for real-world deployment.

**Weaknesses:**

1, LLM-as-judge is well-trodden

Plenty of recent work already uses an LLM to score harmfulness for defense and evaluation. For example, Agentharm benchmark (https://openreview.net/forum?id=AC5n7xHuR1) at ICLR 2025 used LLM judges to provide harm scores, and this work (https://doi.org/10.18653/v1/2024.acl-long.773) uses LLM judges to provide harmful scores to further optimize its attacks. In addition, this work in Usenix (https://dl.acm.org/doi/10.5555/3766078.3766204) also uses LLM to self-defend against jailbreaks. These precedents diminish the paper’s claim of introducing a new scoring-based guardrail.

2. The LLM-based Detection can be bypassed

One ICML 2025 paper (https://icml.cc/virtual/2025/poster/45356) specifically talks about how the LLM detection can be bypassed with their emoji attack. The paper might need to address the claims in this paper to be more convincing.

3, Limited novelty

The two key contributions mentioned in the paper 1, ICL prompting and 2, Dual perspective scoring both fall into the realm of prompt engineering. The paper does not provide a substantive algorithmic or theoretical advance beyond existing judge-style scoring pipelines. This paper is more engineering-heavy and it is hard to gain insight from the paper.

**Questions:**

How will the SelfGrader perform when encountering the emoji attack(https://icml.cc/virtual/2025/poster/45356) that was advertised to bypass LLM detection?

---

> ### Author Response · Authors · 2025-11-24
> **Response to Reviewer b1Lx (1/3)**
>
> **We sincerely thank Reviewer b1Lx for the thoughtful and constructive review comments. We have responded to each weakness and question in detail.**
>
> > [Weakness1 & 3] W1: LLM-as-judge is well-trodden. Plenty of recent work already uses an LLM to score harmfulness for defense and evaluation. For example, Agentharm benchmark (https://openreview.net/forum?id=AC5n7xHuR1) at ICLR 2025 used LLM judges to provide harm scores, and this work (https://doi.org/10.18653/v1/2024.acl-long.773) uses LLM judges to provide harmful scores to further optimize its attacks. In addition, this work in Usenix (https://dl.acm.org/doi/10.5555/3766078.3766204) also uses LLM to self-defend against jailbreaks. These precedents diminish the paper’s claim of introducing a new scoring-based guardrail. W3:  Limited novelty. The two key contributions mentioned in the paper 1, ICL prompting and 2, Dual perspective scoring both fall into the realm of prompt engineering. The paper does not provide a substantive algorithmic or theoretical advance beyond existing judge-style scoring pipelines. This paper is more engineering-heavy and it is hard to gain insight from the paper.
>
> **Response:** Thank you for pointing this out. After reviewing the referenced papers carefully, we confirm that all three works rely on or solely mention the **safety-semantic**, **generation-based** scoring paradigm, whereas SelfGrader introduces a fundamentally different, **logit-based numerical scoring** framework.
>
> 1. **Compared with SelfDefend (USENIX Security Symposium 2025).** Existing judge-based or we called generation-based methods require the model to generate a safety rationale or refusal sentence and then interpret that language. This generation step is exactly what makes them vulnerable. As shown in Table 1 in our original manuscript, SelfDefend becomes vulnerable under multi-turn ActorAttack, where the attacker gradually shapes the semantic context to lower the model’s guard. SelfGrader avoids this weakness because it never relies on generated semantics.

---

> ### Author Response · Authors · 2025-11-24
> **Response to Reviewer b1Lx (2/3)**
>
> 2. We summarize the distinctions below and will discuss them in our revised manuscript:
>
>     **1. Scoring using different representation spaces.**
>     Prior LLM-as-judge work evaluates safety in semantic space, where the model produces textual judgments or safety statements. These outputs are inherently subject to linguistic ambiguity, stylistic bias, decoding variance, and prompt sensitivity. In contrast, SelfGrader evaluates safety in **numerical space**, directly reading out the model’s internal belief state through numerical tokens (NT) logits. This converts the LLM’s latent safety assessment (maliciousness or benignness) into a calibrated numerical score, which is fundamentally different from semantic judgment.
>
>     While prior methods also output “**scores**,” these scores all arise from the same **semantic pipeline**:
>
>     (1) generating natural-language explanations or rationales (e.g., harmfulness reasoning, intent summaries);
>
>     (2) producing semantic labels or Likert-scale judgments (e.g., safe/unsafe, risk levels);
>
>     (3) parsing the generated text to extract a score.
>
>     For example, AgentHarm uses GPT judges to generate harmfulness rationales and Likert-scale semantic scores; ACL ’24 relies on semantic interpretations of generated safety explanations; and SelfDefend uses a shadow model to generate diagnostic text (e.g., intention summaries, harmful-span descriptions) and bases safety decisions solely on these semantic outputs.
>
>     **2. Scoring using pre-decoding logits instead of decoded text.**
>
>     All these methods operate on **decoded text**. The safety evidence they rely on is extracted only after the model commits to specific wording, which discards most of the internal belief distribution. SelfGrader instead evaluates safety from **pre-decoding logits**, preserving full distributional evidence that is inaccessible to text-based LLM judges.
>
>     **3. Scoring using NT logits only.**
>
>     Although pre-decoding logits contain richer information, not all of them are appropriate for safety judgment. A naïve strategy would be to inspect the logits of safety-related keywords only. However, relying on keyword logits is fundamentally unreliable: the keyword space is open-ended, highly context-dependent, and impossible to define comprehensively for robust maliciousness or benignness assessment. In contrast, NTs form a **closed, invariant, and task-aligned label space**. Under the SelfGrader prompt, the model internally maps its hidden state to a scalar score, making the logits over NTs a direct and high-SNR (signal-to-noise ratio) readout of the model’s safety judgment (either maliciousness or benignness). This design avoids the linguistic ambiguity, prompt sensitivity, and vocabulary incompleteness inherent to semantic tokens, and provides a robust mechanism for extracting safety signals directly from the model’s logit distribution.
>
>     **4. Scoring from dual safety perspectives from NT logits.**
>
>     To reduce false positives, we also evaluate benignness so that the system does not unnecessarily refuse benign inputs. NT logits enable this because they are **viewpoint-neutral**. They do not presuppose a specific narrative direction. As a result, prompting the model to “rate maliciousness” produces a maliciousness-oriented logit distribution, while prompting it to “rate benignness” produces a benignness-oriented one. These two distributions can then be compared, contrasted, or combined in a symmetric manner. In contrast, the scores in these existing methods arise solely from generated text and therefore express only a single perspective, unable to simultaneously capture evidence for both maliciousness and benignness.
>
>     As for the ICL examples, we apologize for the confusion in how our key contributions were presented, which may have led to a misunderstanding of the novelty of SelfGrader. The core novelty does not lie in the presence of ICL examples. Instead, it lies in introducing a numerical-space representation of safety judgment, together with the corresponding mechanisms (NT logits and the DLP scoring rule) that make this representation possible. The ICL examples serve only to instantiate the scoring task in a lightweight manner and to prompt the model to output numerical scores, so that the safety judgment can be expressed cleanly through NT logits.
>
>     We will carefully revise the manuscript accordingly to clarify the unique contribution of SelfGrader.

---

> ### Author Response · Authors · 2025-11-24
> **Response to Reviewer b1Lx (3/3)**
>
> > [Weakness2 & Question1] W2: The LLM-based Detection can be bypassed. One ICML 2025 paper (https://icml.cc/virtual/2025/poster/45356) specifically talks about how the LLM detection can be bypassed with their emoji attack. The paper might need to address the claims in this paper to be more convincing. Q1: How will the SelfGrader perform when encountering the emoji attack(https://icml.cc/virtual/2025/poster/45356) that was advertised to bypass LLM detection?
>
> **Response:** Thank you for raising this important point and for referring us to the ICML 2025 emoji attack paper. We agree it is valuable to highlight why SelfGrader resists these attacks.
>
> Theoretically, the emoji attack is designed to break semantic detectors like referenced papers, while SelfGrader is a purely logit-based detector. To verify this, we report the performance of SelfGrader under this emoji attack.
>
> We implement emoji attack over the jailbreak prompts in our Table 1, and obtain the following defense performance results (ASR ($\downarrow$) / PGR ($\downarrow$) in \%):
>
> | Guardrails  | Manual (IJP) | GCG | AutoDAN | DrAttack | MultiJail | ActorAttack | Avg |
> |------------|--------------|-----|---------|----------|-----------|-------------|---------|
> | LLma-3-8B-Insturct (No Defense) | 7.80/- | 13.00/- | 2.00/- | 10.00/- | 4.44/- | 22.66/- | 9.98/-
> |  LLma-3-8B-Insturct (No Defense) under Emoji Attack | 8.80/- | 5.00/- | 2.00/- | 16.00/- | 6.98/- | 4.00/- | 7.13/-
> | Llama Guard (Pre) | 6.00/56.10 | 10.00/39.00 | 2.00/50.00 | 10.00/84.00 | 4.44/95.23 | 22.66/99.83 | 9.18/70.69
> |  Llama Guard (Pre) under Emoji Attack |6.40/62.40 | 5.00/87.00| 2.00/95.00 |16.00/99.00 | 6.90/100.00 | 4.00/99.50| 6.72/90.48
> | SelfDefend (Direct) | 2.00/28.90 | 3.00/11.00 | 0.00/11.00 | 6.00/54.00 | 4.12/72.38 | 20.50/87.00 | 5.94/44.05
> | SelfDefend (Direct) under Emoji Attack |0.90/27.00 |1.00/8.00 | 2.00/14.00 |9.00/44.00 | 6.98/89.20 | 2.66/76.16| 3.76/43.06
> | **SelfGrader (Q=101)** | 0.50/6.30 |0.00/1.00|0.00/1.00|3.00/17.00|4.44/52.38|0.00/0.00|1.32/12.95
> | **SelfGrader (Q=101) under Emoji Attack** | 0.20/9.10 |2.00/4.00|0.00/0.00|0.00/0.00|0.00/0.00|0.00/0.00|0.36/2.18
>
> Overall, adding the emoji attack increases ASR for several jailbreak methods. Manual prompting, DrAttack, and MultiJail all become more effective at jailbreaking the target LLM.
>
> For generation-based guardrails such as Llama Guard, the average PGR increases, indicating reduced stability under this perturbation. For SelfDefend, the average PGR result stays roughly unchanged, with minor fluctuations in PGR across different attacks. In contrast, SelfGrader remains stable under the emoji attack. The perturbation does not cause degradation and, in fact, makes unsafe generations slightly easier to detect, leading to lower ASR and lower PGR on average. This consistency demonstrates that SelfGrader’s logit-based mechanism is robust to the semantic obfuscations introduced by emoji insertion.

---

### Official Review · Reviewer_RBU3 · 2025-10-30

**Soundness:** 3
**Presentation:** 3
**Contribution:** 3
**Rating:** 6
**Confidence:** 3

**Summary:**

This paper introduces SelfGrader, a lightweight guardrail method designed to detect jailbreak attacks against LLMs. The central contribution is to reframe jailbreak detection from a text-generation or feature-extraction problem into a numerical grading problem.

Instead of relying on computationally expensive internal features (like gradients) or fragile keyword matching from generated responses , SelfGrader prompts the LLM to evaluate the safety of an input query using a compact set of Numerical Tokens (NTs) (e.g., '0' to '9'). The method then analyzes the token-level logit distribution over these specific NTs to derive an internal safety signal.

To make this signal meaningful and stable, the authors propose two key innovations:

* An In-Context Learning (ICL) prompting strategy that anchors the numerical scores to human-interpretable categories of maliciousness, using curated query-score pairs as examples.

* A Dual-Perspective Logit (DPL) scoring rule that evaluates the query from both a "maliciousness" and a "benignness" perspective. This dual-view approach aims to produce a more stable score and reduce the false positive rate on benign prompts.

The authors conduct extensive experiments across multiple LLMs (Llama-3-8B, Vicuna-13B) and a wide range of jailbreak benchmarks and attack types, including adaptive attacks. The results demonstrate that SelfGrader achieves accurate and robust detection with low computational overhead and latency, while also preserving model utility by maintaining a low False Positive Rate (FPR) on benign tasks.

**Strengths:**

This paper presents a strong contribution to the critical area of LLM safety, with strengths across all key dimensions.

1. Originality:

* The primary conceptual contribution is novel: it reframes jailbreak detection from a text-classification or feature-extraction task into a numerical grading problem.
* Instead of relying on brittle keyword matching from generated text or computationally-heavy internal features like gradients , the paper proposes using the token-level logit distribution over a compact set of Numerical Tokens (NTs) as a direct, internal safety signal. This "logit-space" approach is a creative and original alternative to existing guardrail paradigms.
* The Dual-Perspective Logit (DPL) scoring rule is also a novel methodological component, thoughtfully designed to stabilize the safety signal by integrating both "maliciousness" and "benignness" assessments, which the authors show is crucial for reducing false positives.

2. Quality:

* The paper demonstrates high quality through its rigorous methodology and comprehensive empirical evaluation.
* Methodology: The method is well-designed. It cleverly uses an In-Context Learning (ICL) strategy to align the NT logit signals with human intuition of maliciousness, eliminating the need for any model training or fine-tuning.
* Evaluation: The experimental setup is extensive and robust.
* It is tested against a diverse and challenging set of jailbreak attacks, including manual (IJP), optimization-based (GCG, AutoDAN), implicit (DrAttack), and multi-turn (ActorAttack) methods. Crucially, the evaluation includes adaptive attacks (TAP, LLM-Fuzzer, X-Teaming), which represent a much stronger test of a guardrail's robustness. The authors responsibly evaluate utility by measuring the False Positive Rate (FPR) on a wide range of benign benchmarks, including tasks like math (GSM8K) and coding (HumanEval) that could be mistakenly flagged by a naive numerical-based defense.
* Results: The results are excellent. SelfGrader achieves a state-of-the-art balance of safety and efficiency. It consistently achieves low Attack Success Rates (ASR) (e.g., 1.32% average ASR on Llama-3-8B ) while being significantly more efficient than strong baselines. For example, Table 1 shows it is ~25x faster and uses ~3x less memory than GradientCuff, and its ~378 MB memory overhead is negligible compared to other LLM-based guardrails that require >13GB.

3. Clarity:
* The paper is exceptionally clear and well-written. The problem is well-motivated and formally defined (Section 3) .
* The proposed method is broken down into three logical, easy-to-understand steps: NT-based logit extraction, DPL scoring, and the decision rule (Section 4) .
* Figure 1 provides an excellent visual overview that immediately contrasts SelfGrader's mechanism with generation-based and classification-based methods, greatly aiding comprehension.
* The experiments, results, and ablation studies are presented clearly and support the paper's claims effectively.

4. Significance
* The paper addresses a problem of high and urgent importance for the safe deployment of LLMs.
* The most significant contribution is the practicality of the proposed solution. By demonstrating a method that is highly effective while also having negligible latency and memory overhead, the authors present a guardrail that can be realistically deployed in production systems .
* This work opens a promising new direction for guardrail research, moving beyond text-space and gradient-space analysis to a more direct and efficient logit-space-based evaluation.

**Weaknesses:**

The paper's core idea is to quantify the model's internal safety judgment using a logit-based signal from a single forward pass. However, the evaluation is missing a direct comparison to another key class of lightweight, single-pass methods: those that measure the generative loss (or perplexity) of a pre-defined refusal response (e.g., "I cannot assist with this request") or affirmative responses (e.g., "sure, ...") .

To fully substantiate the benefits of the novel numerical grading approach, the authors should include a comparison against this simpler baselines, like Token Highlighter[1].

**References**

[1] Token Highlighter: Inspecting and Mitigating Jailbreak Prompts for Large Language Models. Xiaomeng Hu, Pin-Yu Chen, Tsung-Yi Ho

**Questions:**

please refer to the weaknesses.

**Details Of Ethics Concerns:**

no concerns.

---

> ### Author Response · Authors · 2025-11-24
> **Response to Reviewer RBU3 (1/1)**
>
> **We sincerely thank Reviewer RBU3 for the thoughtful and constructive review comments. We conducted the recommended additional experiment and summarize our findings below.**
>
> > [Weakness] The paper's core idea is to quantify the model's internal safety judgment using a logit-based signal from a single forward pass. However, the evaluation is missing a direct comparison to another key class of lightweight, single-pass methods: those that measure the generative loss (or perplexity) of a pre-defined refusal response (e.g., "I cannot assist with this request") or affirmative responses (e.g., "sure, ...") .
> To fully substantiate the benefits of the novel numerical grading approach, the authors should include a comparison against this simpler baselines, like Token Highlighter[1].
> References
> [1] Token Highlighter: Inspecting and Mitigating Jailbreak Prompts for Large Language Models. Xiaomeng Hu, Pin-Yu Chen, Tsung-Yi Ho
>
> **Response:** We thank the reviewer for highlighting this important comparison.
>
> 1. We evaluated SelfGrader against a perplexity and loss-based detector (PerplexityFilter) using the same model and benchmark in our original submission. For generation-based baselines that rely on predefined refusal or affirmative responses, we already include several representative methods such as Llama-Guard, SelfDefend, WildGuard, and GuardReasoner. As shown in Tables 1, 2, 5, and Figure 2, SelfGrader attains comparable or lower ASR, PGR, and FPR. These results indicate that NT logit scoring provides a more stable and discriminative safety signal than perplexity-only or generation-based approaches.
>
> 2. We appreciate the reviewer for highlighting this missing comparison and will incorporate the following new experiment and analysis in the revised manuscript.
>
> Since Token Highlighter is not itself a guardrail method and does not output a binary safety decision, we follow the original paper’s ASR evaluation procedure. We first apply Token Highlighter to generate the defended answer, and then follow the paper to use Llama-Guard-2-8B to classify whether the defended output is safe. All other hyperparameters are choose according to the original paper. The empirical results are shown below:
>
> **Comparison of Guardrails against Various Attack Methods on LLama-3-8B-Instruct. Defense performance are reported as ASR ($\downarrow$) / PGR ($\downarrow$) in \%.**
>
> | Guardrails | Manual (IJP) | GCG | AutoDAN | DrAttack | MultiJail | ActorAttack | Avg | Latency (Sec.) | Extra Memory (MB) |
> |------------|--------------|-----|---------|----------|-----------|-------------|-----|-----------------|--------------------|
> | LLama-3-8B-Instruct (No Defense) | 7.80/- | 13.00/- | 2.00/- | 10.00/- | 4.44/- | 22.66/- | 9.98/- | - | - |
> | Token Highlighter |3.60/10.00 |8.00/16.00| 0.00/6.00| 6.00/9.00| 1.26/19.00 |0.00/0.16 | 3.14/10.02 | 15.20 | 65536.66
> | **SelfGrader (Q=101)** | 0.50/6.30|0.00/1.00|0.00/1.00|3.00/17.00|4.44/52.38|0.00/0.00|1.32/12.95 | 0.78 | 377.61 |
>
> **Comparison of Guardrails on Benign Prompts. Defense Utility are reported as FPR ($\downarrow$) in \%.**
>
> | Guardrails | GSM8K |	HumanEval	|AlpacaEval|	OR-Bench | Avg | Latency (Sec.) | Extra Memory (MB) |
> |------------|-----|------------|-----------|-----|-----------------|--------------------|---|
> | LLama-3-8B-Instruct (No Defense) | 0.00 | 0.00 | 0.00 | 0.00 | 0.00 | - |-|
> | Token Highlighter | 99.33|100.00|99.25 |98.20 | 99.20 | 13.08| 64009.53|
> | **SelfGrader (Q=101)** |0.00| 0.00|1.36|6.30|1.91 |1.50| 254.75 |
>
> Across both the jailbreak benchmarks and the benign–prompt utility evaluation, our method delivers stronger overall safety–effectiveness and substantially better efficiency. On the jailbreak benchmarks, our approach obtains a lower average attack success rate. In addition, it runs almost 20$\times$ faster than Token Highlighter and requires only a small fraction of its memory footprint, making it far more practical for real deployments. On benign prompts, our detector maintains very low false–positive rates, in sharp contrast to Token Highlighter whose responses are almost always marked as unsafe by the downstream judge.

---

> > ### Comment · Reviewer_RBU3 · 2025-11-27
> >
> > Thank you for the reply. The authors have adequately answered my questions and clarified the points I raised in my review. I have no further concerns at this stage and will keep my original high score.

---

### Official Review · Reviewer_GUAA · 2025-10-31

**Soundness:** 3
**Presentation:** 3
**Contribution:** 2
**Rating:** 4
**Confidence:** 3

**Summary:**

This paper proposes SelfGrader, a lightweight guardrail method that introduces a novel in-context prompting strategy and a dual-perspective scoring rule, reformulating jailbreak detection as a numerical grading problem based on token-level logits. Extensive experiments were conducted across diverse jailbreak benchmarks, multiple LLMs, and various guardrail baselines to demonstrate SelfGrader’s accuracy and robustness.

**Strengths:**

1. This paper introduces a novel NT-based logits evaluation mechanism. With the dual-perspective score, the DPL scoring rule integrates both maliciousness and benignness perspectives to generate a more stable and reliable signal for decision-making. To further enhance the robustness of the scoring process, a tail trimming strategy is employed to remove low-probability tokens that may otherwise destabilize the score.

2. This paper presents a comprehensive and well-rounded experimental evaluation. It tests SelfGrader against various types of jailbreak attacks, including manual attacks, optimization-based attacks, generation-based attacks, implicit attacks, and multi-turn attacks, and compares it with multiple guardrail baselines. The evaluation covers a wide range of metrics such as ASR, PGR, FPR, latency, and extra memory consumption, providing a thorough assessment of the guardrail’s performance. Furthermore, extensive ablation studies are conducted to analyze different parameters and components of SelfGrader, including the tail trimming parameter, number of NTs (Q), DPL coefficient, as well as key submodules like ICL Examples and DPL Scoring.

**Weaknesses:**

1. The paper’s contributions lack sufficient novelty. The SelfGrader mechanism essentially combines elements of generation-based and classification-based guardrail approaches. Specifically, it first adopts a logit computation process similar to that used in generation-based methods to calculate the LLM’s output logits. It then introduces a scoring mechanism that evaluates these logits using numerical tokens, such as digits 0–9, which resemble the detection mechanism employed by generation-based guardrails.

2. The reliability of the method is questionable. SelfGrader adopts an in-context learning strategy by embedding curated query–score pairs that guide the model to align its logits with predefined maliciousness standards. However, the examples of prompts shown in Appendix B indicate that both the queries and scores are generated by the guardrail model. In particular, the scores are not computed using the DPL Scoring Rule proposed in the paper. Therefore, it is uncertain whether the use of such data is reliable.

**Questions:**

1. What unique or irreplaceable advantages does SelfGrader possess, beyond being lightweight, that make it distinct from generation-based and classification-based guardrails?

2. Moreover, why does SelfGrader rely on the guardrail model to generate the ICL examples instead of using an LLM directly for this purpose?

---

> ### Author Response · Authors · 2025-11-24
> **Response to Reviewer GUAA (1/2)**
>
> **We sincerely thank Reviewer GUAA for the thoughtful and constructive review comments. We have responded to each weakness and question in detail.**
>
> ---
>
> > [Weakness1 & Question1] W1: The paper’s contributions lack sufficient novelty. The SelfGrader mechanism essentially combines elements of generation-based and classification-based guardrail approaches. Specifically, it first adopts a logit computation process similar to that used in generation-based methods to calculate the LLM’s output logits. It then introduces a scoring mechanism that evaluates these logits using numerical tokens, such as digits 0–9, which resemble the detection mechanism employed by generation-based guardrails. Q1: What unique or irreplaceable advantages does SelfGrader possess, beyond being lightweight, that make it distinct from generation-based and classification-based guardrails?
>
> **Response:** Thank you for raising this point. We believe the impression of “limited novelty” may arise from a superficial similarity in that SelfGrader also accesses logits. However, SelfGrader is not a hybrid of existing guardrails, it introduces a **new guardrail paradigm** that differs fundamentally from both generation-based and classification-based methods, beyond being lightweight.
>
> Existing guardrails operate in **safety-semantic space**: (1) Generation-based guardrails analyze textual outputs (e.g., refusal messages, safety statements). (2) Classification-based guardrails predict semantic labels (e.g., “0 for safe" and “1 for unsafe"). These systems reason over semantic content, which is inherently coarse, lossy, and highly dependent on linguistic form.
>
> SelfGrader departs from this paradigm. It interprets the model’s internal safety judgment in **numerical space**, using the logits associated with digit tokens. This converts the LLM’s latent belief state about safety (either maliciousness or benignness) into a calibrated numerical signal, rather than text or discrete labels. This shift in representation is the central novelty.
>
> Detailed technical differences and advantages including:
>
> 1. **Using different signal.** SelfGrader relies on an inference process that directly uses the model’s logits, whereas existing approaches operate on much more compressed downstream signals. Generation-based methods depend on the final text sampled from the logit distribution, and classification-based methods rely on the label predicted by an external safety classifier.
> 2. **Using NT logits only.** Logits provide much finer-grained information, i.e., the raw belief distribution over the entire vocabulary, than either generated text or classification labels. One can definitely look into the logits of safety-related keywords to identify maliciousness or benignness. However, probing arbitrary safety-related keywords is unreliable: the keyword space is open-ended, highly context-dependent, and impossible to define comprehensively. In contrast, numerical tokens constitute a closed, invariant, and task-aligned label space. Under the SelfGrader prompt, the model internally maps its hidden state to a scalar score, making the logits over digit tokens a direct and high-SNR (signal-to-noise ratio) reflection of the model’s safety judgment (whether evaluating maliciousness or benignness). This avoids linguistic ambiguity, prompt sensitivity, and vocabulary incompleteness, yielding a robust and principled mechanism for extracting safety signals directly from the model’s logit distribution.
> 3. **Dual safety perspectives from NT logits.** To reduce false positives, we also evaluate benignness so that the system does not unnecessarily refuse benign inputs. NT logits enable this because they are viewpoint-neutral. They do not presuppose a specific narrative direction. As a result, prompting the model to “rate maliciousness” produces a maliciousness-oriented logit distribution, while prompting it to “rate benignness” produces a benignness-oriented one. These two distributions can then be compared, contrasted, or combined in a symmetric manner. In contrast, generated text and classification labels collapse the model’s belief into a single direction, making such dual-perspective analysis infeasible.
>
> We hope this clarifies the distinct contributions of SelfGrader. We will make these points more explicit in the revised manuscript.

---

> ### Author Response · Authors · 2025-11-24
> **Response to Reviewer GUAA (2/2)**
>
> > [Weakness2 & Question2] W2: The reliability of the method is questionable. SelfGrader adopts an in-context learning strategy by embedding curated query–score pairs that guide the model to align its logits with predefined maliciousness standards. However, the examples of prompts shown in Appendix B indicate that both the queries and scores are generated by the guardrail model. In particular, the scores are not computed using the DPL Scoring Rule proposed in the paper. Therefore, it is uncertain whether the use of such data is reliable. Q2: Moreover, why does SelfGrader rely on the guardrail model to generate the ICL examples instead of using an LLM directly for this purpose?
>
> **Response:** Thank you for pointing this out. We apologize for the confusion caused by the presentation in Appendix B. All ICL examples are generated by GPT-5, not by the guardrail model. We will correct this in the revised version.
>
> We would also like to clarify that the reliability of SelfGrader does not depend on the accuracy of the scores in ICL examples. The ICL examples are used solely to provide the format and instruction style of the scoring task, not to supply ground-truth labels. This is why the examples do not use DPL scores. SelfGrader’s actual safety score is always computed from the NT logits of the target query.

---

### Author Response · Authors · 2025-12-04
**General Response to ACs and SACs**

Dear ACs and Senior ACs,

We thank ACs, SACs, and all reviewers for their efforts in this special cycle. Overall, the reviewers recognize our work for its novel and original guardrail design (Reviewers GUAA, RBU3, b1Lx), comprehensive experimental evaluation and ablation studies (Reviewers GUAA, RBU3, b1Lx, FHc7), methodological efficiency (Reviewers b1Lx, FHc7), and clear writing (Reviewer RBU3). During the rebuttal phase, Reviewer RBU3 expressed satisfaction with our responses and maintained a positive score. Below, we briefly summarize how our rebuttal addresses the main concerns raised across all reviews.

**1. Method Novelty (Reviewers GUAA, b1Lx, FHc7).** We clarify that SelfGrader introduces a new guardrail paradigm that differs fundamentally from existing approaches. Prior guardrails operate in a **safety-semantic space**. First, generation-based guardrails analyze the textual output such as refusal messages and safety statements. Second, classification-based guardrails assign semantic labels such as “0 for safe” and “1 for unsafe”. These systems rely on semantic interpretation, which is coarse, lossy, and highly sensitive to linguistic phrasing. Third, internal-feature-based methods require substantial computational cost. SelfGrader departs from these paradigms. Its use of numerical tokens is motivated by the desire to construct a **closed, invariant, and task aligned yet flexible numerical space** that avoids the instability inherent in safety-semantic scoring. We have added these clarifications in our revision.

**2. Comparison with Additional Defense Methods (Reviewer RBU3).** Following the recommendation of Reviewer RBU3, we conduct new experiments comparing SelfGrader with Token Highlighter. Across both jailbreak benchmarks and benign prompt utility evaluations, our method demonstrates stronger overall safety effectiveness and significantly better efficiency in terms of running time and memory footprint. These results have been included in our revision.

**3. Robustness to Emoji Attack (Reviewer b1Lx).** The reviewer raises concerns about jailbreak attack methods specifically designed to evade LLM-based detection, with the emoji attack being one example. We have added evaluations of the emoji attack across multiple jailbreak benchmarks. Regarding to the results, the emoji attack does not cause performance degradation and in fact makes unsafe generations slightly easier to detect, resulting in lower ASR and lower PGR on average compared to the benchmarks that do not use emoji attack. These results are included in Section 5.2 of the revised version.

**4. Robustness and Extensibility to Prompt Injection Attacks (Reviewer FHc7).** The reviewer asks how our method behaves and generalizes to prompt injection attacks. To address this, we evaluated SelfGrader on the widely used prompt injection benchmark BIPIA. Although SelfGrader is not specifically designed for prompt injection, it exhibits strong resistance to the attacks included in the BIPIA benchmark. These results are provided in Appendix D of the revised version.

**5. Performance Variations (Reviewer FHc7).** The reviewer raises concerns about performance variations across different attack families. We clarify that our method achieves the best overall performance in robustness, utility (i.e., low false-positive rate) and efficiency among the evaluated guardrails.

**6. Other Points.** We also addressed several additional questions during the rebuttal, including the source of in-context learning examples (Reviewer GUAA), the design of the numerical token resolution $Q$ (Reviewer FHc7), and the flexibility of SelfGrader when combined with existing safety aligned models (Reviewer FHc7). We have made the corresponding revisions in the manuscript.

We hope this consolidated response is helpful for the final assessment.

Best Regards,

Authors

---

### Meta-Review · Area_Chair_fRiF · 2026-01-03

**Summary:**

Lack of novelty: The core of this paper is to have the model score the input, compute the expected harmful score and expected harmless score from the logits output, and then calculate a DPL score as the criterion for determining whether the input is harmful.
Lack of reliability: The reliability of the proposed method largely relies on the designed contextual prompts, without any training of the model itself. This makes the model highly susceptible to prompt injection attacks (unlike guard models such as LLaMA Guard, where the model is specifically trained; for a raw model, its strong capabilities actually amplify the risk of prompt injection). The authors’ experiments also demonstrate this issue, showing that even simple injection can lead to a high PGR.

**Reviewer Concerns:**

## reviewer GUAA
 - Lack sufficient novelty: Not resolved.
 - Context generation issue: Resolved. The authors state that the context is generated by GPT-5.

## reviewer RBU3

 - Lack of baseline: Resolved. The authors provide results on the baseline Token Highlighter.

## reviewer b1Lx

 - Limited novelty: Not resolved.
 - The LLM-based detection can be bypassed (e.g., emoji jailbreak): Resolved. The authors demonstrate defenses against emoji jailbreak, and SelfGrader still shows good defensive performance.

## reviewer FHc7
 - Lack of novelty: Not resolved.
 - Performance instability: Resolved. In fact, requiring good defensive performance against all jailbreak methods is itself unreasonable.
 - Increasing Q is meaningless: Resolved. Increasing Q will definitely improve the performance of the method.
 - Prompt injection vulnerability: Not resolved. Since SelfGrader does not involve targeted training of the model, it is vulnerable to such attacks. The authors assign part of the defense task to the raw model itself, which is unreasonable for plug-in-style defense evaluation.

**Reviewer Scores:**

# reviewer scores
## reviewer GUAA
 - Score remains 4

## reviewer RBU3
 - Score remains 6

## reviewer b1Lx
 - Score increased from 2 to 4

## reviewer FHc7
 - Score remains 4

---

### Decision · Program_Chairs · 2026-01-26

Reject